# The Effects of a Mediterranean Diet on Metabolic Hormones and Cytokines in Amyotrophic Lateral Sclerosis Patients: A Prospective Interventional Study

**DOI:** 10.3390/nu17091437

**Published:** 2025-04-25

**Authors:** Anca Moțățăianu, Ion Bogdan Mănescu, Georgiana Șerban, Valentin Ion, Rodica Bălașa, Sebastian Andone

**Affiliations:** 1Department of Neurology, University of Medicine, Pharmacy, Science and Technology of Târgu Mureș ‘George Emil Palade’, 540142 Târgu Mureș, Romania; 21st Neurology Clinic, Mures County Clinical Emergency Hospital, 540136 Târgu Mureș, Romania; 3Department of Laboratory Medicine, University of Medicine, Pharmacy, Science and Technology of Târgu Mureș ‘George Emil Palade’, 540142 Târgu Mureș, Romania; 4Doctoral School, University of Medicine, Pharmacy, Science and Technology of Târgu Mureș ‘George Emil Palade’, 540142 Târgu Mureș, Romania; 5Faculty of Pharmacy, Department of Analytical Chemistry and Drug Analysis, University of Medicine, Pharmacy, Science and Technology of Târgu Mureș ‘George Emil Palade’, 540142 Târgu Mureș, Romania; 6Drug Testing Laboratory, University of Medicine, Pharmacy, Science and Technology of Târgu Mureș ‘George Emil Palade’, 540142 Târgu Mureș, Romania

**Keywords:** amyotrophic lateral sclerosis, ALS, Mediterranean diet, ALSFRS, cytokines, metabolic hormones

## Abstract

**Background**: Amyotrophic lateral sclerosis (ALS) is a prevalent neurodegenerative disease but lacks effective treatments. Dietary interventions, notably the Mediterranean diet, promise to modulate disease pathways. This study aimed to investigate the impact of the Mediterranean diet on gut hormones and cytokines in patients with amyotrophic lateral sclerosis (ALS). **Methods**: We conducted a 12-month, single-center prospective study on a total of 44 ALS patients. After a 6-month observation period, the patients were placed on a dairy-free Mediterranean diet for the next 6 months. We evaluated the patients at baseline (T0), 6 months (T1), and 12 months (T2). We measured the ALS Functional Rating Scale—Revised (ALSFRS-R) scores and a panel of metabolic hormones and cytokines. **Results**: The ALSFRS-R scores declined over 12 months (37.59 ± 6.32 at T0 vs. 30.23 ± 8.91 at T2, *p* < 0.001), indicating expected disease progression with no significant difference in the rate of decline before and after the dietary intervention. The leptin levels significantly decreased from T0 to T1 (T0: 4956 ± 3994 pg/mL vs. T1: 3196 ± 2807 pg/mL, *p* = 0.038). The insulin and GLP-1 levels showed significant drops at T2 (insulin T0: 480 ± 369 vs. T2: 214 ± 213 pmol/L, *p* < 0.01; GLP-1 T0: 118 ± 76 vs. T2: 60 ± 57 pg/mL, *p* < 0.01). C-peptide increased at T2 (T0: 3814 ± 1967 vs. T2: 9532 ± 4000 pg/mL, *p* < 0.001). Among the cytokines, the levels of IL-12P70, IL-13, IL-9, and IL-2 significantly decreased from T0 to T2 (all *p* < 0.05), while IL-17A and TNFα significantly increased between T1 and T2 (*p* < 0.01). **Conclusions**: The Mediterranean diet intervention in ALS patients modulated several metabolic hormones and cytokines but with no evidence of impacting the disease’s evolution or of a slowed clinical progression. These findings suggest a potential role for dietary intervention, particularly the Mediterranean diet, in modulating gut hormones and cytokines in ALS patients, but its impact on disease course is unclear. Future randomized studies are needed to confirm these changes and to determine whether dietary intervention can have any benefit in ALS.

## 1. Introduction

Amyotrophic lateral sclerosis (ALS) is the most common neurodegenerative disease among young and middle-aged adults, with an unknown etiology and a lifetime risk of 1 in 400. This devastating condition is characterized by progressive motor neuron loss, resulting in muscle weakness, paralysis, and a median survival of 36 months. Despite significant efforts in ALS research, it remains a challenging medical condition with limited treatment options, and effective disease-modifying treatments remain elusive, underscoring the urgent need for novel therapeutic strategies [1,2].

Dietary interventions have garnered attention in ALS research due to their potential to modulate the metabolic pathways and inflammatory processes implicated in the neurodegenerative disease’s pathogenesis [3,4]. The Mediterranean diet has emerged as a front-runner due to its rich bioactive compounds with potential neuroprotective and anti-inflammatory properties. Characterized by high consumption of fruits, vegetables, legumes, whole grains, fish, and olive oil, the Mediterranean diet has been associated with a reduced risk of chronic diseases and improved overall health outcomes [5,6]. For example, a Mediterranean-like intervention improved cognitive function in patients at risk for dementia [7].

Metabolic hormones, including C-peptide, active ghrelin, and GLP-1, play crucial roles in energy metabolism, insulin regulation, and neuroprotection [8,9,10]. Dysregulation of these hormones has been implicated in ALS progression, highlighting their potential as targets for therapeutic intervention. Concurrently/together, cytokines, such as IL-2, IL-6, and TNFα, orchestrate immune responses and inflammatory processes implicated in neuroinflammation and motor neuron degeneration characteristic of ALS [11,12,13].

Our study aimed to investigate the impact of the Mediterranean diet intervention on metabolic hormone levels and cytokine profiles in patients with ALS. We evaluated changes over 12 months at three time points to elucidate the potential associations between dietary intervention and the relevant humoral biomarkers of ALS pathophysiology. We meticulously analyzed longitudinal data to explain how the Mediterranean diet intervention influences these biomarkers, shedding light on its therapeutic efficacy and mechanisms in ALS management. Our study aimed to investigate the impact of the Mediterranean diet on metabolic hormone and cytokine levels in patients with ALS over a 12-month period. We present here the longitudinal results, focusing on changes in key metabolic hormones (e.g., C-peptide, ghrelin, GLP-1) and cytokines (e.g., IL-2, IL-13, IL-17A, TNFα) associated with the dietary intervention. Understanding the effects of dietary intervention on these biomarkers could offer valuable insights into their therapeutic potential in ALS management. Through a comprehensive assessment and analysis, we aim to contribute to the growing evidence supporting dietary interventions as adjuvant therapies for ALS to improve ALS prognosis.

## 2. Materials and Methods

### 2.1. Study Design and Participant Selection

This prospective, interventional, single-center study was designed to evaluate the impact of dietary intervention on the progression of ALS. Initially, 44 patients were enrolled during the baseline visit (T0) between August 2022 and November 2022 from those diagnosed with ALS in the Neurology Department of the Mures University Clinical Emergency Hospital. We longitudinally assessed the ALS patients according to the study protocol for 12 months, with 3 visits at 6-month intervals. After the first six months (T1 visit), only 36 patients survived, and the diet intervention was initiated. After six months of diet, ultimately, only 30 ALS patients survived until T2 (final evaluation visit). The surviving ALS patients underwent follow-up assessments every six months at our department over 12 months. The flow chart of the study design can be observed in Figure 1.

The target sample size of approximately 40 ALS patients was chosen based on similarities to previous studies in ALS [14,15] given the rarity of the disease and the anticipated dropout rate. According to this, our final sample of 30 patients corresponds with sample sizes reported in comparable ALS interventional studies.

All the ALS patients were diagnosed by experienced clinical ALS neurologists, and the participants were exclusively drawn from the Romanian population residing in Mures County, ensuring homogeneity in racial demographics. The inclusion criteria stipulated the following: (1) fulfillment of the El Escorial–Airlie House criteria for definite or probable ALS diagnosis (with possible ALS cases excluded), as well as compliance with the Awaji criteria [16,17]; (2) age 25 years or older; (3) ability to adhere to study protocols; (4) absence of familial ALS history; and (5) attendance of three clinic visits at 6-month intervals for surviving ALS patients. The exclusion criteria comprised the following: (1) diagnosis of other neurodegenerative diseases; (2) presence of neurological conditions or comorbidities potentially interfering with ALS progression assessment; (3) history of diabetes mellitus or gastrointestinal tract diseases and medications, including gastrointestinal surgeries; (4) immune-suppressive therapy within the preceding six months.

Additionally, all the ALS patients received standard ALS therapy, including 100 mg/day of riluzole. All the patients signed the informed consent before the first visit.

### 2.2. Diet Intervention

The nutritional approach for ALS patients involved adhering to a dairy-free Mediterranean diet, with a macronutrient distribution comprising approximately 50% carbohydrates from healthy cereal sources (whole grains, fruits, and vegetables), approximately 30% fat primarily from plant-based sources (such as flaxseed, nuts, seeds, and olive oil), and small fatty fish (such as sardines, mackerel, herring, and salmon) due to their creatine content, which plays a role in ATP production and potentially enhances muscular performance in ALS patients. Approximately 20% of the protein intake came from lean organic meat sources (excluding red meat), eggs, and fish. Emphasis was placed on incorporating prebiotic foods like leeks, garlic, oats, artichokes, and apples to support a balanced gut microbiome. The diet also prioritized micronutrients, particularly essential fatty acids such as alpha-linolenic acid, eicosapentaenoic acid (EPA), and docosahexaenoic acid (DHA), sourced from both marine and plant-based foods, polyphenols and flavonoids, fruits and vegetables. Additionally, antioxidant-rich foods were recommended [5,6].

### 2.3. Data Collection

Baseline demographic and clinical data regarding ALS were collected, including ALS duration (time from initial symptom presentation to first clinical evaluation) and a comprehensive neurological assessment, incorporating the ALS Functional Rating Scale—Revised (ALSFRS-R), with calculated functional subscores for each domain: bulbar, lower limb, upper limb, and respiratory [18]. The rate of ALSFRS-R progression (ΔPR) was calculated as the decline in ALSFRS-R score from diagnosis to the date of sample collection using the formula: 48 − [(ALSFRS-R at diagnosis − ALSFRS-R at study visit)/duration of symptoms between onset and study visit (in months)] [19]. Each visit was documented separately.

The survival period was recorded. In the event of a participant’s demise, the date of death was acquired from local registration offices. At the baseline visit (T0), 44 patients were included; 36 patients remained at T1, and only 30 ALS patients survived at T2. All the participants provided informed consent before the baseline visit (T0).

The study protocol was approved by the Ethics Committee of the Clinical County Emergency Hospital Mures (Approval No. 13472/21.06.2022) and by the University Ethics Board of UMFST “George Emil Palade” (Approval No. 1841/21.07.2022). All the participants provided written consent prior to inclusion. This research strictly adhered to pertinent guidelines and regulations, aligning with the principles of the Declaration of Helsinki.

### 2.4. Biological Specimen Collection

Blood samples were collected from each ALS patient at every study visit (T0, T1, T2) in the morning after an overnight fast, using EDTA tubes. Immediately after collection, a protease inhibitor cocktail was added, and the samples were centrifuged. The plasma supernatant was aliquoted into 0.5 mL vials and stored at −80 °C until analysis.

### 2.5. Measurement of Metabolic Hormones and Cytokines via Multiplex Bead-Based Immunoassay

Serum samples, previously stored at −80 °C, were transported on ice to the Humoral Immunology Laboratory of the Center for Advanced Medical and Pharmaceutical Research (Targu Mures, Romania). Upon thawing, the samples underwent vigorous vortexing and were processed using a commercially available multiplex bead-based immunoassay, following the manufacturer’s instructions (MILLIPLEX^®^ Human Th17 Magnetic Bead Panel kit; Merck-Millipore catalog number HTH17MAG-14K, Merck KGaA, Darmstadt, Germany). The assay simultaneously measured the following 25 cytokines: CCL20/MIP3α (chemokine C-C motif ligand 20/macrophage inflammatory protein 3), GM-CSF (granulocyte-macrophage colony-stimulating factor), IFNγ (interferon gamma), interleukin (IL)-1β, IL-2, IL-4, IL-5, IL-6, IL-9, IL-10, IL-12P70, IL-13, IL-15, IL-17A, IL17E/IL-25, IL-17F, IL-21, IL-22, IL-23, IL-27, IL-28A, IL-31, IL-33, TNFα (tumor necrosis factor alpha), and TNFβ.

The MILLIPLEX^®^ Human Metabolic Hormone Magnetic Bead Panel kit (Merck-Millipore catalog number HMHEMAG-34K, Merck KGaA, Darmstadt, Germany) was used for the metabolic hormone assay: total amylin, insulin, C-peptide, GIP (gastric inhibitory peptide), active ghrelin, GLP-1 active (glucagon-like peptide-1), glucagon, PP (pancreatic polypeptide), PYY (peptide YY), leptine, MCP-1 (monocyte chemoattractant protein-1), and TNFα (tumor necrosis factor alpha). Subsequently, the samples were analyzed utilizing Luminex ^®^ xMAP ^®^ technology, employing a properly calibrated and controlled FLEXMAP 3D ^®^ analyzer (Luminex Corporation, Austin, TX, USA), with data analysis conducted using xPONENT ^®^ software version 4.3.

The results are expressed in ng/mL or pg/mL, depending on each parameter.

### 2.6. Statistical Analysis

Statistical analyses were performed using IBM SPSS Statistics v26 and Microsoft Excel 2019.

Parametric variables were assessed through ANOVA testing using a Bonferroni correction, with continuous data described using mean and standard deviation (SD).

We used the Shapiro–Wilk test to assess the normality of our data distributions.

We used a combination of Microsoft Excel 2019 and Adobe Photoshop CS4 for graphical figures.

## 3. Results

### 3.1. General Population Characteristics

Our studied groups contained 44 ALS patients aged 58.39 ± 12.52 in T0, of whom 36 remained alive for T1. In the end, only 30 patients were alive in T2 with an age of 57.43 ± 12.39.

We observed a few significant differences regarding the ALSFRS-R subscores between the three visits (all the *p*-values reported are after Bonferroni correction).

The respiratory and gross motor subscores differed significantly between T0 and T2 but not between T1 and the other two.

The bulbar subscore had no significant differences between any of the three visits.

The fine motor subscore and total score had significant differences between T0 and T1 and T0 and T2 but no difference between T1 and T2.

No differences were observed regarding the ΔPR progression rate between the three visits.

The complete data can be found in Table 1.

### 3.2. Metabolic Hormone Analysis

The evaluation points (T0, T1, T2) showed a few differences in serum metabolic hormone values.

Leptin had a decrease in value from T0 to T1 (*p* = 0.038) but no other statistical differences regarding the values between T1 and T2 or T0 and T2 (Anova test) (Figure 2A).

Insulin has a decrease in serum value from T0 to T2 (*p* = 0.009) and from T1 to T2 (*p* = 0.042), without any statistically significant difference between T0 and T1 (Anova test) (Figure 2B).

GLP-1 serum values were statistically significant differences between T0 and T2 (*p* = 0.001) and between T1 and T2 (*p* = 0.002) but were not significant between T0 and T1 (Anova test) (Figure 2C).

MCP-1 values increased from T0 to T1 (*p* = 0.009) and decreased from T1 to T2 (*p* = 0.008). However, the difference between T0 and T2 was not statistically significant (Anova test) (Figure 2D).

C-peptide serum values at T2 were significantly higher as compared with those at both T0 (*p* < 0.001) and T1 (*p* < 0.001). However, no significant differences were found between T0 and T1 (Anova test) (Figure 2E).

PP serum levels had no statistical differences between T0 and T1 but had statistical differences between T0 and T2 (*p* = 0.034) and T1 and T2 (*p* = 0.003) (Anova test) (Figure 2F).

All the differences between the serum values of metabolic hormones can be found in Appendix A.

### 3.3. Cytokine Analysis

IL-12P70 values were significantly lower in T2 than in T0 (*p* = 0.005), with no differences regarding T1 values to either point (Anova test) (Figure 3A).

IL-13 serum values were significantly lower in T2 than in T0 (*p* = 0.026), while T1 values showed a slight increase in average value compared with that at T0; this difference had no statistical significance (Anova test) (Figure 3B).

IL-17A decreased in value from T0 to T1 but with no significance, followed by a sudden increase in value in T2. The difference between T1 and T2 was extremely significant (*p* < 0.001), even if the one between T0 and T2 was not (Anova test) (Figure 3C).

IL-9 followed a downward trend from T0 to T1 and to T2, but the only statistically significant difference was between T0 and T2 (*p* = 0.047) (Anova test) (Figure 3D).

IL-2, as well, followed a similar trend; however, the differences between T1 and T2 and between T0 and T2 were both statistically significant (*p* = 0.022 and *p* = 0.004, respectively) (Anova test) (Figure 3E).

TNFa decreased in value from T0 to T1 but without significance, followed by a rise in value in T2. The only statistically significant difference was between T1 and T2 (*p* = 0.002) (Anova test) (Figure 3F).

IL-22 values decreased from T0 to T2 (*p* = 0.009) and from T1 to T2 (*p* = 0.020), but there were no statistically significant differences between T0 and T1 (Anova test).

All the other cytokines showed no statistically significant differences between any of the three points in time. The complete data are available in Appendix A.

## 4. Discussion

Our study shows evidence that the Mediterranean diet can modulate metabolic hormones and inflammatory cytokines in ALS patients over a 6-month intervention. We observed significant changes in multiple metabolic hormones: C-peptide serum levels increased significantly after the diet, while insulin levels decreased. Active ghrelin levels almost doubled after 6 months of diet, and the leptin levels declined. In terms of inflammatory cytokines, we observed that pro-inflammatory IL-12p70, as well as IL-9 and IL-2, but also anti-inflammatory IL-13, all showed significant reductions following the diet. By contrast, IL-17A and TNF-α serum levels increased during the same period.

C-peptide, a peptide hormone synthesized in pancreatic beta cells during proinsulin processing, has long been regarded solely as a byproduct of insulin production. However, recent studies propose that C-peptide may possess independent biological effects beyond its role in insulin secretion. Unlike insulin, C-peptide has a longer half-life and greater stability, making it a reliable marker of insulin secretion [20]. Moreover, emerging research suggests potential neuroprotective properties of C-peptide in neurodegenerative diseases such as ALS. At physiological levels, C-peptide exhibits recognized anti-inflammatory, anti-apoptotic, and antioxidant functions, which may impede underlying processes involved in ALS pathogenesis.

The study by Aydemir et al. offers valuable insights into the systemic changes occurring during the progression of ALS in a rat model, shedding light on potential biomarkers crucial for early disease detection and monitoring. Their findings reveal notable alterations in C-peptide levels, among other biomarkers, indicating a complex metabolic dysregulation characteristic of ALS. Interestingly, C-peptide levels exhibit an initial increase during the early phase of the disease, followed by a significant decrease in the symptomatic stage. This dynamic pattern underscores the potential utility of C-peptide as a diagnostic and prognostic indicator in ALS [21].

Ghrelin, a gut hormone known for regulating appetite and energy balance, also exerts neuroprotective effects through multifaceted mechanisms. Ghrelin’s ability to influence both metabolic and neurological processes underscores its significance in maintaining overall health and suggests promising avenues for therapeutic intervention in conditions such as neurodegeneration [22,23,24]. Nagaoka et al. have demonstrated that ghrelin could be used as an independent predictor marker for ALS survival, and ghrelin’s alterations lead to faster ALS progression and poorer survival [25]. A prominent feature in the development of ALS is neuronal damage triggered by the accumulation of activated microglia in affected areas. This accumulation releases pro-inflammatory molecules and reactive oxygen and nitrogen species, contributing to neuroinflammation.

Recent studies, such as those conducted by Howe et al., have revealed a connection between ALS and impaired ghrelin release. These findings suggest that altered ghrelin levels may contribute to the metabolic changes observed in ALS and influence disease progression. Lower levels of ghrelin have been associated with specific disease characteristics, decreased body mass, and potentially worse survival outcomes in ALS patients [14].

Based on research findings indicating the potential benefits of ghrelin supplementation in ALS, several publications advocate for its use as a potential therapeutic approach to improve ALS prognosis. Ghrelin supplementation has been suggested for its neuroprotective effects, preservation of muscle mass, and regulation of metabolism, and it is likely to improve survival outcomes in ALS patients [15,26]. Our study revealed a twofold increase in ghrelin levels at T2 compared with those at T0 and T1, indicating the potential of a controlled diet in regulating the nutritional status of ALS patients, thereby avoiding malnutrition and weight loss. However, we found no statistically significant correlations with ALS severity scores. The observed increase in ghrelin levels after six months of following a Mediterranean diet in ALS patients likely reflects a combination of factors, including improved nutritional status, dietary composition, anti-inflammatory effects, and modulation of gut microbiota. Further research is warranted to fully comprehend this observation’s mechanisms and implications for ALS management.

Leptin, an appetite-suppressant hormone derived from adipocytes, and ghrelin, a gut hormone known for its orexigenic properties, are pivotal in regulating appetite and energy balance. Often seen as “opposing” hormones, their dysregulation has been linked to several neurological disorders, including ALS. Imbalances in leptin and ghrelin signaling may disrupt metabolic homeostasis and potentially impact the progression of ALS [27,28,29]. Concurrently with the weight loss commonly observed in ALS, the circulating levels of leptin, an adipose tissue-derived hormone, are reduced in ALS patients. However, elevated levels of leptin have been associated with a reduced risk of developing ALS and, in those already diagnosed, with improved survival outcomes [30].

The detrimental impact of the downregulation of the glucagon-like peptide-1 (GLP-1) pathway in ALS is well established, exacerbating disease progression through various mechanisms, including impaired neuroprotection, metabolic dysregulation, mitochondrial dysfunction, excitotoxicity, and neuroinflammation [31]. Moreover, research underscores the significant influence of downregulated IGF-1 and GLP-1 signaling pathways on ALS progression, with dysregulation of these pathways contributing to neurodegeneration. Activation of GLP-1 offers the potential for restoring abnormal signaling, promoting neuronal regeneration, and providing neuroprotection and neurotrophic effects in ALS [32]. Likewise, Diz-Chaves et al. endorse using GLP-1 receptor agonists as a promising therapeutic strategy, emphasizing their ability to evoke an anti-inflammatory response in neurodegenerative disorders [33].

Surprisingly, both GLP1 and GIP levels decreased at T2 compared with those at T0 and T1, indicating that dietary implementation does not augment incretin levels. Further comprehension of the GLP-1 pathway in ALS pathophysiology could provide valuable insights into potential therapeutic interventions targeting this pathway to attenuate disease progression and mitigate symptoms.

MCP-1, or monocyte chemoattractant protein-1, serves as a chemokine responsible for recruiting and activating monocytes and other immune cells at sites of inflammation. In the context of ALS, MCP-1 is implicated in neuroinflammation, marked by elevated levels of pro-inflammatory cytokines and chemokines [34]. The initial rise in MCP-1 levels from T0 to T1 may be attributed to disease progression.

In our study, the rise in MCP-1 values from T0 to T1 may indicate the inflammatory response during the natural progression of ALS without dietary intervention. However, the subsequent decrease after six months of the Mediterranean diet intervention suggests reduced inflammation associated with dietary changes. The anti-inflammatory properties of the Mediterranean diet, coupled with potential enhancements in metabolic health and modulation of gut microbiota, likely contribute to the observed fluctuations in MCP-1 levels throughout the study period. Additionally, we observed significant negative correlations between MCP-1 levels and ALSFRS-R scores (specifically respiratory and bulbar). These associations underscore the detrimental impact of MCP-1 on ALS prognosis, highlighting its potential as a prognostic marker in disease progression. Zhai et al. demonstrated that supplementing the diet with short-chain fatty acids, particularly butyrate, reduces hepatic expression of MCP1/CCL2, resulting in an ensuing anti-inflammatory effect [35]. Following the implementation of our dietary intervention, MCP1 levels in ALS patients decreased, confirming previous findings.

Amylin, a satiation hormone secreted by pancreatic beta-cell islets, plays a crucial role in reducing the expression of orexigenic neuropeptides and is also implicated in the formation of amyloid plaques not only in the pancreas but also in the central nervous system (CNS) [36]. Epidemiological studies indicate that amylin may be implicated in the intricate interplay between metabolic disturbances such as type 2 diabetes mellitus (T2DM) and neurodegenerative diseases, notably Alzheimer’s disease (AD) [36].

In our study, we observed fluctuations in amylin levels throughout the study period. Initially, there was a decrease in amylin levels in the first six months. Interestingly, after implementing the Mediterranean diet intervention at T2, there was a notable increase in amylin levels. These findings suggest that dietary changes may influence amylin secretion in ALS patients, warranting further investigation into the role of amylin in ALS pathophysiology and its potential as a therapeutic target.

Our study followed the longitudinal changes in both pro- and anti-inflammatory cytokines over one year following the diagnosis of ALS. The first six months captured the natural course of the disease, while the subsequent six months assessed the impact of dietary interventions on disease progression. The existing scientific literature has yielded conflicting results due to the heterogeneous phenotype and genotype of ALS patients and the influence of disease stages on cytokine levels [12,37,38,39].

Among the pro-inflammatory cytokines studied (IL-10, IL-13, IL-4, and IL-5), only IL-13 showed a statistically significant difference. IL-13 is typically present in healthy brain tissue and is upregulated in response to cerebral injury to protect damaged areas. Elevated IL-13 levels have been associated with induced apoptosis of activated microglia, reduced sensitivity to excitotoxicity-induced neuronal death (a recognized pathway in ALS pathogenesis), and enhanced synaptic plasticity [40]. Previous research has linked IL-13 to a better prognosis in neurodegenerative disorders like AD by promoting macrophage uptake of beta-amyloid and reducing amyloid plaque formation in the CNS [41].

Our findings revealed similar IL-13 levels between the initial assessment (T0) and six months later (T1) but a significant decrease at the one-year mark (T2) after six months of dietary intervention than at the time of diagnosis. This suggests that adherence to a Mediterranean diet may not improve the level of IL-13 and probably the prognosis in ALS patients. However, evidence regarding the influence of the Mediterranean diet on IL-13 levels is lacking, highlighting the need for further research to confirm or infirm its potential benefits.

IL-2 is a well-known immunoregulatory cytokine that plays a crucial role in promoting both the initiation and resolution of inflammatory immune responses [41]. In neurodegenerative diseases such as ALS, IL-2 exhibits dual effects. It enhances the development and function of regulatory T cells, potentially exerting a neuroprotective and anti-inflammatory role, particularly in the early stages of slowly progressive disease. Conversely, IL-2 can activate natural killer cells in end-stage disease, leading to cytotoxic effects on neurons [39,41].

Studies have shown conflicting results regarding the relationship between IL-2 levels and ALS progression. Moreno-Martinez et al. found a significant correlation between increased IL-2 levels, faster ALS progression, and shorter overall survival [42]. Similarly, Lu et al. suggested that elevated IL-2 might be a risk factor for survival in ALS patients [38]. However, the exact role of IL-2 in ALS pathogenesis remains unclear. Ehrhart et al. proposed that IL-2 may have reduced involvement in the initial stages of ALS, but, over a 6-month natural disease progression, IL-2 levels decreased significantly compared with those in healthy controls, indicating a dynamic relationship between IL-2 and disease stages [39].

Our study revealed a continuous decrease in IL-2 levels, particularly pronounced after dietary intervention. However, due to the contrasting effects of IL-2 on immune system activity and the conflicting results in the previous literature, it is challenging to determine the precise influence of the diet on IL-2 levels. Although statistical differences were observed between T2 and T1 and between T2 and T0, further research is necessary to elucidate the impact of diet on IL-2 release in ALS patients.

TNF-α is a well-known pro-inflammatory cytokine implicated in the exacerbation of cognitive and motor impairments in neurodegenerative disorders, including ALS. There is a consensus that TNF-α levels increase with the progression of ALS, with more pronounced effects observed in the later stages of the disease [38]. Our study corroborates these findings, showing that adherence to a Mediterranean diet decreases pro-inflammatory TNF-α levels in ALS patients.

IL-12p70 is a crucial immunoregulatory cytokine that is primarily produced by antigen-presenting cells, influencing both innate and adaptive immune responses [43]. Multiple studies have consistently reported elevated levels of this cytokine in ALS patients than in healthy individuals, along with a positive correlation between IL-12p70 plasma concentrations and overall survival [44,45]. Research by Amrousy et al. demonstrated a reduction in IL-12p70 levels three months after implementing a Mediterranean diet in patients with inflammatory bowel disease [46]. However, Urpi-Sarda et al. investigated the long-term effect of the Mediterranean diet on pro-inflammatory cytokine levels in cardiovascular disease patients and did not find a beneficial impact on IL-12p70 levels after three years [47]. Our study observed a significant decrease in IL-12p70 plasma concentration six months after initiating a Mediterranean diet in ALS patients compared with that at the time of diagnosis, supporting the short-term findings of the Amrousy et al. study [46]. However, further longitudinal monitoring of patients is necessary to validate or infirm the longer-term results reported by the Urpi-Sarda et al. study [47].

Neuroinflammation plays an essential role in the onset and progression of neurodegenerative disorders like ALS, with the Th17/IL-17A pathway emerging as a significant player in ALS pathogenesis [48]. Th17, a subset of T helper cells, produces IL-17A, a cytokine known for its ability to exacerbate neuroinflammation by activating microglia and recruiting other immune cells, ultimately leading to neuron damage [48,49]. Elevated levels of IL-17A in the blood have been associated with ALS [44]. Additionally, the IL-17A concentration is influenced by the composition of the gut microbiota [48].

Studies have shown that individuals following a Mediterranean diet tend to have reduced levels of IL-17A [50]. However, our findings contradict these previous results. We observed that ALS patients adhering to a Mediterranean diet experienced an increase in IL-17A levels at T2 compared with those at T1, although the difference between T0 and T2 was not statistically significant. Further research is warranted to clarify the relationship between Mediterranean diet adherence and IL-17A levels in ALS.

IL-9 is a cytokine known for its regulatory role in modulating immune system pathways, balancing detrimental and protective effects [51]. Despite previous investigations into the impact of the Mediterranean diet on IL-9 levels in various conditions such as ischemic heart disease and T2DM yielding no significant findings, our study revealed a statistically significant reduction in IL-9 levels at T2 compared with those at T0 [52]. However, further research is warranted to elucidate the precise mechanism by which IL-9 is involved in ALS pathogenesis and its potential role in predicting disease progression or developing an IL-9-targeted therapy.

The interventional study on ALS patients over one year revealed notable trends in metabolic hormones and cytokines, shedding light on potential therapeutic strategies and disease progression markers. Regarding metabolic hormones, significant differences were observed in C-peptide and active ghrelin levels, with a marked increase at T2 compared with those at baseline. GLP-1 levels significantly decreased at T2, indicating potential dietary influences on these metabolic markers. Cytokine analysis identified interesting patterns, particularly in IL-12P70, IL-13, IL-17A, IL-9, IL-2, and TNF-α. IL-12P70 and IL-13 decreased significantly at T2, while IL-17A significantly increased, suggesting a dynamic immune response over the study period. IL-9 and IL-2 demonstrated downward trends, with significant reduction at T2, indicating potential associations with disease progression. TNF-α levels increased significantly at T2, indicating intense inflammation.

Notably, the Mediterranean diet in other populations has been associated with decreased levels of inflammatory biomarkers like IL-6 and C-reactive protein [53]. Similarly, in our ALS cohort, we observed significant lower levels in certain cytokines (IL-12p70, IL-13, IL-9, IL-2) following the diet, although pro-inflammatory IL-17A and TNF-α serum levels increased, underscoring the complex immune response of this intervention.

The increase in pro-inflammatory cytokines (like IL-17A and TNF-α) and decrease in anti-inflammatory cytokines (IL-13) during the initiation of the Mediterranean diet in ALS patients may seem counterintuitive, given the diet’s generally recognized anti-inflammatory properties. However, several factors could contribute to this finding: (1) Individual variability, different individuals may respond differently to dietary interventions due to variations in genetics, microbiome composition and others factors. (2) Complex interactions, because the Mediterranean diet is rich in antioxidants and omega-3 fatty acids with anti-inflammatory effects, but specific components of the diet or their interaction with individual biology may trigger immune responses in some individuals, leading to increased cytokine production. (3) Microbiome modulation, because the Mediterranean diet can influence gut microbiota composition, which plays a crucial role in immune system regulation, potentially leading to an increased production of pro-inflammatory cytokines in some individuals. (4) Adaptive responses, with an initial increase in pro-inflammatory cytokines may reflect the immune system’s adaptive response to dietary changes and this transient increase may be part of regulatory mechanism aimed at restoring homeostasis in the face of dietary modifications. (5) Underlying disease mechanisms, because ALS itself involves complex immune dysregulation, and the effects of dietary interventions on immune function may interact with underlying disease mechanisms. In some cases, the observed increase in pro-inflammatory cytokines may reflect the progression of ALS pathology rather than a direct effect of the diet [54,55,56,57].

### Limitations

Several limitations of this study should be acknowledged. First, the sample size was relatively small and drawn from a single center, which may limit the generalizability of the findings. Also, there was a decrease in the sample size, due to the normal disease evolution, during follow-up (only 30 of 44 patients survived to T2), which could introduce bias, as those with more aggressive disease may have been lost. Additionally, this study was not blinded, and all the patients received the diet (no randomization), so placebo effects or other biases cannot be excluded. Finally, multiple biomarkers were tested in an exploratory manner; although we applied Bonferroni corrections, this study may not have been powered to detect small changes for every marker. These limitations suggest caution in interpreting the results and indicate the need for larger, controlled studies to confirm our observations.

## 5. Conclusions

In summary, a Mediterranean diet intervention in ALS patients modulated several metabolic hormones and inflammatory cytokines, revealing a complex interaction between nutrition and ALS pathophysiology. However, we did not observe any significant slowing of clinical progression in our cohort. These findings provide knowledge regarding the biological effects of dietary modulation in ALS, but their clinical significance remains uncertain.

Future studies can confirm these metabolic and cytokines changes and determine whether diet can slow down the disease progression or improve clinical outcome. Our results are the foundation for such trials and suggest that metabolic and immune pathways could be important targets in ALS management.

## Figures and Tables

**Figure 1 nutrients-17-01437-f001:**
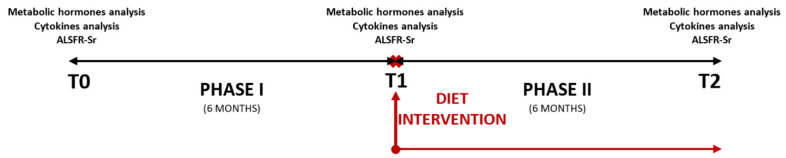
Study design flow chart.

**Figure 2 nutrients-17-01437-f002:**
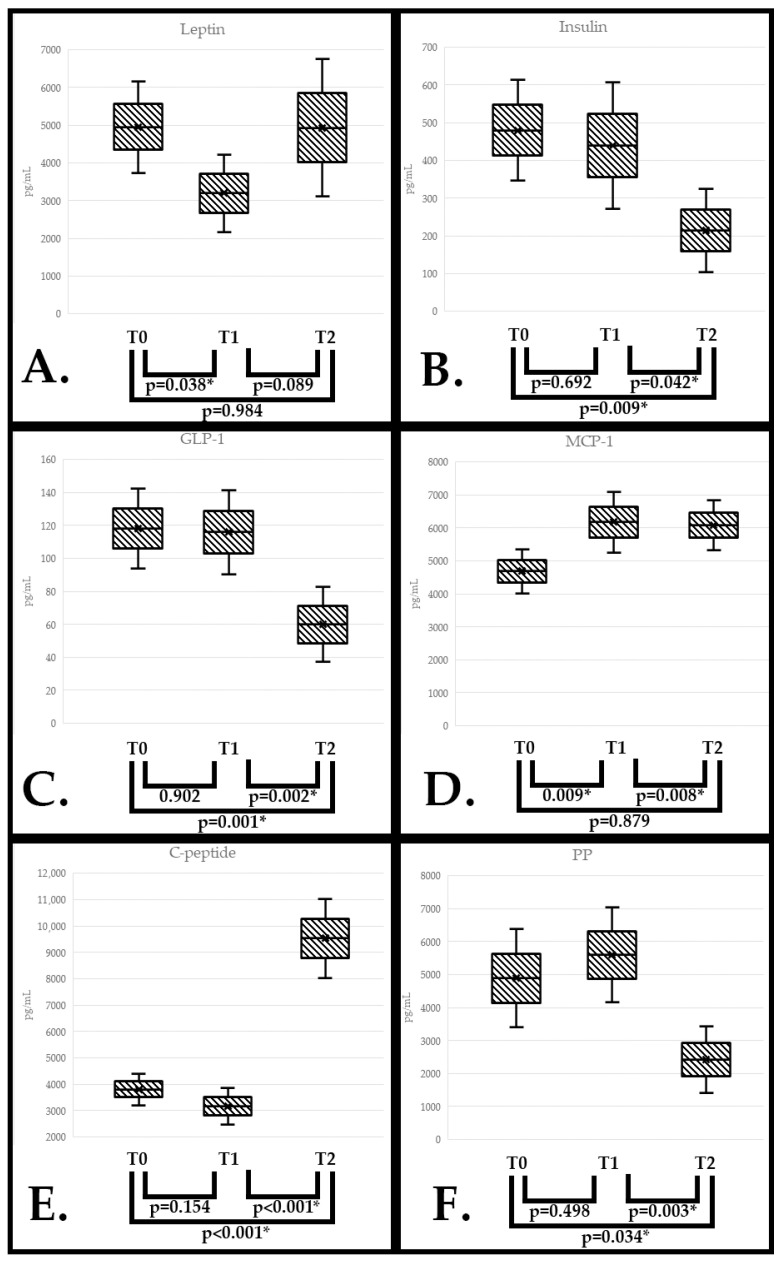
Serum value differences between the 3 visits for leptin (**A**), insulin (**B**), GLP-1 (**C**), and MCP-1 (**D**), C-peptide (**E**), PP (**F**). Data show the interquartile range (IQR), with the lower and upper edges representing the first and third quartiles, respectively, while the whiskers extend to the lower and upper bounds (CI 95%). All the *p*-values reported are after Bonferroni correction. * = statistical significance.

**Figure 3 nutrients-17-01437-f003:**
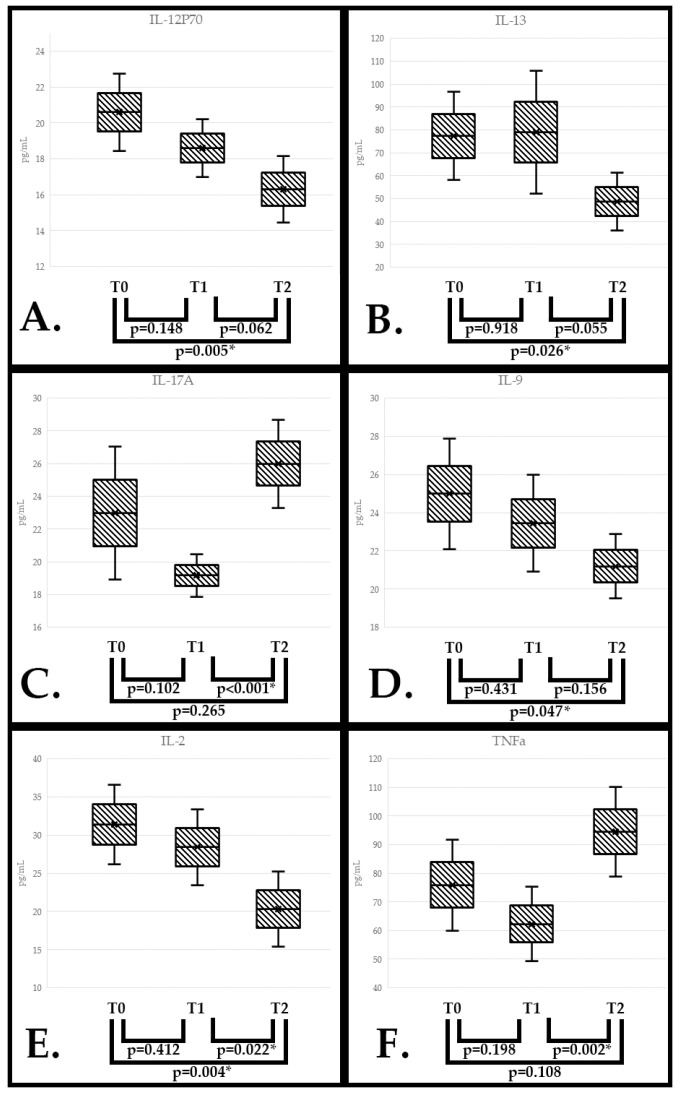
Serum value differences between the 3 visits for IL-12P70 (**A**), IL-13 (**B**), IL-17A (**C**), IL-9 (**D**), IL-2 (**E**), TNFα (**F**). Data show the interquartile range (IQR), with the lower and upper edges representing the first and third quartiles, respectively, while the whiskers extend to the lower and upper bounds (CI 95%). All the *p*-values reported are after Bonferroni correction. * = statistical significance.

**Table 1 nutrients-17-01437-t001:** Patients’ clinical characteristics.

Variable	T0 (*n* = 44)	T1 (*n* = 36)	T2 (*n* = 30)	T0 vs. T1	T1 vs. T2	T0 vs. T2
Age	58.39 ± 12.52	58.17 ± 11.78	57.43 ± 12.39	0.657	0.804	0.746
Sex ratio (M:F)	28:16	23:13	20:10	-	-	-
ΔPR	0.92 ± 1.05	0.75 ± 0.67	0.54 ± 0.34	0.412	0.124	0.061
**ALSFRS-R**						
Respiratory subscore	11.41 ± 0.92	11 ± 1.33	10.23 ± 2.34	0.110	0.100	0.004 *
Bulbar subscore	10.09 ± 2.36	9.39 ± 2.72	9.27 ± 2.89	0.220	0.860	0.182
Gross motor subscore	7.34 ± 3.15	6.03 ± 3.19	5.53 ± 3.14	0.069	0.530	0.018 *
Fine motor subscore	8.75 ± 2.68	6.53 ± 3.39	5.87 ± 3.66	0.002 *	0.450	0.000 *
Total score	37.59 ± 6.32	32.69 ± 7.01	30.23 ± 8.91	0.002 *	0.214	0.000 *

* = statistical significance.

## Data Availability

The data presented in this study are available on request from the corresponding author but will not be publicly available due to restrictions from our funding contract.

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
