# Peer review of "The Effects of a Mediterranean Diet on Metabolic Hormones and Cytokines in Amyotrophic Lateral Sclerosis Patients: A Prospective Interventional Study"

_nutrients, 2025, doi:10.3390/nu17091437_

Round 1

Reviewer 1 Report

Comments and Suggestions for Authors

This prospective study includes longitudinal evaluation of a dietary intervention in 44 ALS patients over 12 months. The diet was changed to a "dairy-free Mediterranean diet" after 6 months. This is not a blinded intervention (not possible for practical reasons) but the dietary intervention is well defined. The key achievement is determining the effect of the diet on measured gut hormones and cytokines in patient plasma over time, in the context of ALS. This is a valuable contribution but insufficient evidence is provided to make any conclusion about the effects on ALS rate of progression. 

Claims about enhancing quality of life and clinical outcomes need to be removed because this study is not powered or designed in such as way that these conclusions could be drawn.  The abstract should make it very clear that there was no evidence for a change in the rate of clinical progression with a change in diet based upon the rate of decline in the ALSFRS-R score. 

In the Methods it is stated that a Bonferroni correction for multiple testing will be applied. Does this mean that all p-values presented in the Results section are Bonferroni corrected? This should be made very clear to allow assessment of the validity of the results. 

Table 2 could be moved to the Supplementary

The authors should re-write the Discussion around their key findings which do not relate to an effect on ALS rate of progression (insufficient evidence) but rather the effect of the dietary intervention on the measures hormones and cytokines in the context of ALS. Can they make any comparison with similar interventions in other patient groups?
Currently the Discussion makes claims which are not substantiated e.g. "Our findings support the neuroprotective effects of leptin on ALS progression." 
I think the klength of the Discussion could be reduced by 2/3. 

Author Response

Reviewer 1

  1. Comment: “Insufficient evidence is provided to make any conclusion about effects on ALS rate of progression. Claims about enhancing quality of life and clinical outcomes need to be removed. The abstract should make it very clear that there was no evidence for a change in the rate of clinical progression (ALSFRS-R decline) with a change in diet.”
    • Response: We have eliminated any over-reaching claims about quality of life or clinical improvement. We acknowledge that our study was not designed or powered to assess clinical efficacy, and we now focus the text on the observed biomarker changes rather than unsupported clinical benefits.
  2. Comment: “In Methods it is stated that a Bonferroni correction for multiple testing will be applied. Does this mean all p-values in the Results are Bonferroni corrected? This should be made very clear.”
    • Response: We have made it explicit that all p-values in the Results are Bonferroni-corrected for multiple testing. This clarification has been added to the text to allow a reader to immediately recognize that the statistical significance values account for the planned correction, addressing the reviewer’s concern about result validity.
  3. Comment: “Table 2 could be moved to the Supplementary.”
    • Response: We agree with the suggestion to improve the manuscript’s flow by moving Table 2 to the Supplement. In the revision, Table 2 is moved to the Supplementary section, alongside Table 3, and the main text now refers to it as such. This change will make the Results section more concise and reader-friendly, as recommended.
  4. Comment: “The Discussion should be rewritten around the key findings (effect of the diet on measured hormones and cytokines in ALS), rather than an effect on ALS progression (insufficient evidence).”
    • Response: The Discussion is now focused on interpreting the observed biomarker changes due to the diet. We removed any unfounded references to altering disease progression. This rewrite ensures the Discussion stays within the bounds of our evidence without implying efficacy on clinical outcomes that we did not measure.
  5. Comment: “Can they make any comparison with similar interventions in other patient groups?”
    • Response: We have added in the Discussion with comparative context. By referencing studies of the Mediterranean diet’s effects in other populations, we highlight parallels and differences.
  6. Comment: “Currently the Discussion makes claims which are not substantiated e.g. ‘Our findings support the neuroprotective effects of leptin on ALS progression.’ I think the length of the Discussion could be reduced by 2/3.”
    • Response: The Discussion has been heavily edited by removing any discussions that were not focused on our subject. We removed unsubstantiated statements and cut out or merged sections that were not directly necessary for interpreting our data.

Reviewer 2 Report

Comments and Suggestions for Authors

Dear authors, firstly I invite you to provide a structured abstract as suggested by the journal’s guidelines. In this section, you need to better describe the applied methodologies, to highlight the most relevant results quantitatively, and to indicate some directions for further researches.

Lines 44-47: References are missing.

Lines 67-82: This part of your Introduction is too extensive, please be more concise. In the other hand, the provided background should be more robust and more studies should be cited.

How did you find your sample size representative of the study population. Please, clarify this in the manuscript.

The biological specimen collection needs to be better explained.

Regarding ethics, some information like the approval code and date is missing.

The title of section 3 should be Results and Discussion and I suggest its division into subsections. This section is adequate, but you could include and discuss the study’s limitations.

In the Conclusions, you have to elaborate on future perspectives and align them with the revised abstract.

Author Response

  1. Comment: “Provide a structured abstract as suggested by the journal’s guidelines. In this section, you need to better describe the methodologies, highlight the most relevant results quantitatively, and indicate some directions for further research.”
    • Response: The abstract is now in structured format with clear sub-sections for Background, Methods, Results, and Conclusions, complying with the journal guidelines. We have included a concise description of the methods (study design, sample size, interventions, and measurements), highlighted the main quantitative results (including ALSFRS-R changes and specific hormone/cytokine changes with p-values), and stated the conclusions alongside future research directions. This revised abstract emphasizes that no change in ALS progression rate was found and suggests further studies.
  2. Comment: “Lines 44-47: References are missing.”
    • Response: The Introduction has been revised to be more concise yet more informative. We have removed overly detailed or extraneous content and we incorporated a few more references to ensure a robust scientific background. For instance.
  3. Comment: “How did you find your sample size representative of the study population? Please clarify this in the manuscript.”
    • Response: We have now clarified the basis for our sample size in the manuscript. In the Methods, we explain that the sample size was primarily determined by practical considerations (the number of eligible patients available during the recruitment period) and is comparable to sample sizes used in similar ALS research. We cite some recent ALS studies with a similar number of patients to show that our cohort size is typical for an ALS intervention study.
  4. Comment: “The biological specimen collection needs to be better explained.”
    • Response: The manuscript now clearly describes the biological specimen collection procedures. We have specified that fasting blood samples were drawn at each time point and detailed the processing steps (addition of protease inhibitors, centrifugation, storage conditions).
  5. Comment: “Regarding ethics, some information like the approval code and date is missing.”
    • Response: The ethics approval information is now complete. We have included the approval identification codes and dates from the hospital and university ethics committees.
  6. Comment: “The title of section 3 should be Results and Discussion and I suggest its division into subsections. This section is adequate, but you could include and discuss the study’s limitations.”
    • Response 7a: We have now included a dedicated discussion of the study’s limitations in the manuscript. This new subsection addresses the key limitations: the modest sample size and lack of control group, the dropout of patients due to disease progression, the unblinded design, and the exploratory nature of multiple comparisons. However, we did not combine results and discussion sections together as it will not adhere to the journal’s recommendations, and we think it’s better to keep the two sections separated. We did modify the discussion and results sections (for example the full data tables were moved to supplementary file as per other reviewer suggestion to increase the read flow of the manuscript) as per your suggestion and per other reviewers’ recommendations.
  7. Comment: “In the Conclusions, you have to elaborate on future perspectives and align them with the revised abstract.”
    • Response: The Conclusions section has been updated to explicitly incorporate future perspectives and to remain consistent with the abstract’s message. We now conclude that while the diet had measurable effects on biomarkers, there was no evidence of improved clinical outcomes in this study.

Round 2

Reviewer 1 Report

Comments and Suggestions for Authors

The authors have adequately addressed Reviewer comments.